# Automated Learning of Radiation Therapy Dose Distribution for Head and Neck Cancer

**Davood Hajinezhad**[*]                                      davood.hajinezhad@sas.com
**Afshin Oroojlooy**[*]                                         afshin.oroojlooy@sas.com
**Xin Hunt**[*]                                                          xin.hunt@sas.com
**Mohammadreza Nazari**[*]                                    reza.nazari@sas.com
**Jorge Silva**[*]                                                   jorge.silva@sas.com
[*] *SAS Institute Inc.*

**Shiva Das**[†]                                            shivadas@email.unc.edu
[†] *University of North Carolina at Chapel Hill*

**Editors:** Under Review for MIDL 2021

## Abstract

In the past decades, 3D dose optimization, with the goal of providing a prescription dose to the tumor while maximally sparing surrounding healthy organs at risk (OARs), has been playing a pivotal role in external beam radiation therapy. To reach a practical clinical treatment plan, the optimization solution needs to be adjusted by human planners. However, this procedure necessitates intensive computations combined with iterative inputs from an expert human planner. This is a time-intensive process with dose distribution solutions that are highly dependent on the human expertise of an institution. To remedy the computational time issue, and also reduce the variability, while capturing the existing knowledge, we use machine learning capabilities. In particular, we propose a suitable feature matrix for head and neck cancer and train Deep Neural Network models to mimic the Planned Dose Distribution procedure. When trained, the model can provide dose values in less than a second. Toward this, a specific Convolutional Neural Network called U-Net model is utilized. The numerical results show that this model reduces the solution time of each patient on the utilized data from 3-6 hours to 1 second while providing highly accurate solutions such that the average error is approximately 3.2% and 7.0% for cancerous organs and OARs, respectively.

**Keywords:** Radiotherapy, Machine Learning, U-Net

## 1. Introduction

Radiotherapy is a very common treatment for cancer, which utilizes radiation to destroy cancerous tissues. This type of treatment is built upon the paradigm that cancerous tissue is preferentially eliminated by giving it higher therapeutic radiation doses compared to surrounding healthy tissues. Thus, the challenge is to provide sufficient doses to cancerous tissues while maximally sparing surrounding healthy organs at risk (OARs) from collateral damage. When a patient is diagnosed with cancer and physicians decide to include radiation therapy as part of the treatment plan, the most frequent mode of delivery of radiation dose is via either Intensity Modulated Radiation Therapy (IMRT) (Webb, 2001; Cedric et al., 2002) or Volume Modulated Arc Therapy (VMAT) (Otto, 2008). Both of these methods

deliver radiation using a linear accelerator, where the radiation beam is created and then directed at the tumor (the beam essentially travels through the healthy tissues to reach the tumor). Instead of delivering the radiation using a single beam, these methods irradiate beams in multiple orientations, thereby providing a higher dose at the intersection (tumor) compared to surrounding OARs.

The mathematical models for IMRT and VMAT are very complex and hard to solve as they are usually modeled as mixed integer or constrained nonlinear programming (MINLP) models (Ehrgott et al., 2010). These optimization models have significantly improved radiotherapy planning quality. Nonetheless, all of them suffer from intensive computational requirements. In addition to the computationally-intensive optimization, in real-life patient treatment planning, the process is interactive whereby the optimization constraints and weights are continuously manually adjusted by a human planner (over the course of the optimization) to achieve a balanced trade-off between the collateral doses to OARs versus the target. Consequently, the treatment 3D Planned Dose Distribution (PDD) for a specific patient is a result of the combination of the optimization algorithm and human manipulation (interactive adjustments of optimization constraints and weights). In essence, this 3D PDD is almost impossible to replicate by a fully automated optimization algorithm because of the large and important extent of human contribution. The need for multiple interactive adjustments, combined with the long optimization time required by the software, can lead to delays in the treatment planning phase. These delays in the treatment course may cause detrimental consequences for the patients as well as the demand for additional resources from the health care providers such as hospitals.

Additionally, since the process of producing a PDD is highly dependent on the planner's experience, expertise, or preference, different physicians may obtain distinct treatment plans. Therefore, deployed software packages are not a perfect choice for producing clinically acceptable plans among physicians as they are not incorporating the best practices of human experts in their solutions. This procedure is illustrated in Figure 1 (a). In this figure, the *Patient Information* contains prescribed dose values for cancerous tissues, limit dose values, cancerous tissues, and OARs. In this workflow, the optimization block itself imposes expensive computations and it is the main bottleneck of the treatment planning. Since the human expert may need to re-run the optimization many times, shortening the optimization time can improve the procedure significantly. This motivates us to harness the power of Machine Learning models to generate plans that are of high quality regardless of physician while reducing processing time for obtaining a dose distribution pattern.

Deep Neural Network (DNN) is an information processing system and one of its main characteristics is the capability to *learn* from data. Thanks to the significant improvements in computational powers such as GPUs and parallel computations on one hand, and the ever-increasing amount of digitally stored data on the other hand, DNNs have gained outstanding attention in many areas in recent years. At the same time, the advancements in radiation oncology procedures such as CT-Scan and MRI, and the shift to using electronic health records (EHR) result in a considerable amount of data that is being generated and stored. As a result, DNNs and their variants such as Convolutional Neural Networks (CNN) (LeCun et al., 1995) and U-Net (Ronneberger et al., 2015) have been used and provided promising results in a wide range of medical applications such as image segmentation, outcome prediction, dose quantification, and radiation adaptation.

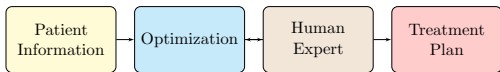

(a) PDD procedure contains an optimization algorithm followed by a human expert to adjust the dose values during an iterative interaction. This plan is computationally expensive and the solutions are not robust.

(b) In the proposed workflow, a DNN model incorporates the knowledge from the PDD plan for training. After training, this model is able to provide a more robust guidance for the human expert toward designing the final treatment plan.

Figure 1: The workflow for the PDD and proposed DNN-based model

In this work, we propose a DNN-based method with a special input matrix to address the main shortcomings of PDD: (i) the costly computations and (ii) the variability of the solutions. To address these issues, we incorporate previous human knowledge in the optimization process to train a DNN model over the PDD outputs. Figure 1 (b) presents our proposed workflow to obtain the treatment plan. Once we have a trained DNN, the model provides a solution very quickly. This solution would be a significant guidance for human experts to provide the final plan. Furthermore, unlike the PDD, less variability is expected on the solutions due to the fact that DNN model is trained over a variety of human adjustments. Also, we expect that the number of interactions between the machine and the human expert would be lower than that of the optimization. In this work, our proposed model is trained over existing PDD plans from 100 patients with head and neck cancer. Several performance measurements is conducted which verify the effectiveness of the proposed method. Specifically, our numerical experiments suggest that our proposed framework reduces the solution time to less than 1 second (vs. a few hours required to solve it with the optimization within the current approach) with about [1.5%, 9.8%] error in PDD.

## 2. Related Work

Due to the importance of radiotherapy in cancer treatment, a broad range of researchers contributed in this area. Linear and nonlinear programming approaches were developed in (Rosen et al., 1991) and (Källman et al., 1992) respectively. To improve the treatment plan using optimized aperture shapes, column generation approaches were utilized in (Romeijn et al., 2005; Mahnam et al., 2017). In (Aleman et al., 2010) an interior point method was introduced to optimize the radiation intensity for each beam. In (Christiansen et al., 2018), a continuous aperture fluence calculation was proposed for VMAT process. A heuristic algorithm based on a mixed-integer linear programming model was proposed in (Dursun et al., 2019), where the dose requirements are considered as conditional value-at-risk constraints. To accelerate the optimization algorithms, a sampling strategy was proposed in (Yan et al., 2018) for the beam arrangement of VMAT planning. To see more optimization related algorithms, the readers are referred to the review article in (Unkelbach et al., 2015).

With the recent progress in Machine Learning and advancements in Deep Learning, radiotherapy gained remarkable attention from these communities as well. In (Kann et al., 2018) a CNN model was proposed for identification of nodal metastasis and tumor extra-nodal extension. Similarly, in (Nikolov et al., 2018), a U-Net is proposed to automate the delineation of radio-sensitive OARs. In (Sadeghnejad Barkousaraie et al., 2020) U-Net is proposed to mimic a Column Generation optimization algorithm for *beam orientation*

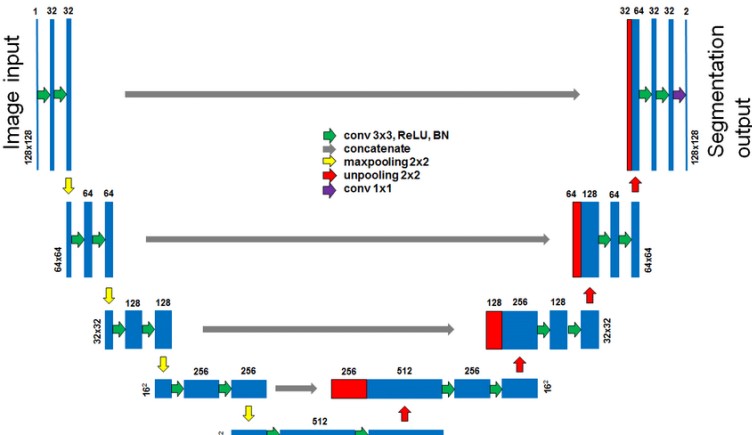

Figure 2: The left side of this U-shape model is the contracting part, which consists of two copies of two convolutional layers with kernel size $3 \times 3$, and one max-pooling layer with kernel size $2 \times 2$. The bottom part is called bottleneck and uses two $3 \times 3$ CNN layers followed by $2 \times 2$ up-convolution layer. The expanding part, which is the right side, consists of repeated blocks each containing two convolutional layers followed by an up-convolutional.

problem. Another popular paradigm for predicting dose distribution is called Knowledge-based Planning, in which the Dose Volume Histogram (DVH) of a patient is predicted using historical information. Despite the large improvements that Knowledge-based Planning brought in radiotherapy, these methods suffer from huge feature manipulation to be successful. To alleviate this complexity, in (Nguyen et al., 2019), a DNN model was trained to produce dose distributions for prostate cancer based on existing Knowledge-based Planning solutions. See more relevant works in (Boldrini et al., 2019).

## 3. The Proposed Model

In this section, we elaborate the proposed model for predicting the dose distribution in radiotherapy.

### 3.1. U-Net Model

The U-Net model includes a sequence of convolution, max-pooling, and transposed convolution operations in order to extract local and global features and to return the image to the original size. See Fig 2. In particular, U-Net consists of two main blocks, namely the *contracting block* and the *expanding block*. The left side of this U-shape model is the contracting part, which consists of two copies of two convolutional layers with kernel size 3, and one max-pooling layer with kernel size $2 \times 2$. The expanding part, which is the right side of the U-shape, consists of repeated blocks each containing two convolutional layers followed by an *up-convolutional* layer. All convolutional layers have used the ReLU activation function.

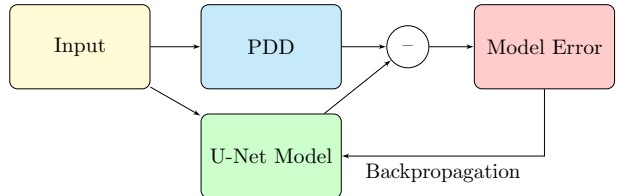

Figure 3: The U-Net model gets the prescribed dose values, type of tissue, dose limits for OARs, and distance to target tissue as the input to U-Net and fits the model to predict the PDD from the treatment plan.

### 3.2. Data

Treatment plans gained using PDD is available for 100 anonymized patients at the university of North Carolina at Chapel Hill (UNC) school of medicine. In this data, the set $\texttt{OARs} = \{\texttt{Brain-Stem, Cord, Larynx, Right-Parotid, Left-Parotid}\}$ is considered as the OARs. Also, cancerous structures (targets) are also categorized into *planning target value with standard risk* and *planning target value with high risk*, which are denoted by $\texttt{PTV-SR}$, and $\texttt{PTV-HR}$, respectively. Therefore, we denote the tissue set as $T = \{\texttt{OARs, PTV-SR, PTV-HR}\}$. For every patient, there exists (i) a set of $Z$ images of size $512 \times 512$ ($Z$ denotes the depth of the CT scan image which depends on the patient length), (ii) the prescribed dose values to target, and (iii) the limit dose values for OARs. Also, the treatment PDD provides a tensor of size $512 \times 512 \times Z$ containing the dose values for each pixel of the body.

### 3.3. Training U-Net Model

For the input, we define feature space consisting of a four-channel image: (i) the prescribed dose values for target, (ii) the type of tissue from the set $T$, (iii) the dose limit value for OARs, and (iv) the distance to target for each pixel which is the Euclidean distance to the closest target tissue. Therefore, we have a $512 \times 512 \times 4$ tensor for each $z = 1, \ldots, Z$. Since the prescribed values are only available for the target, we need to estimate the corresponding values for OARs. To this end, we fit an exponential function that estimates the incidental dose values based on the distance of each organ from the nearest target pixel. We performed several ablation studies on the feature representation and noticed that removing any of these features causes the loss of accuracy in the prediction model. For the labels, we simply consider the PDD from the treatment plan, which is a tensor of size $512 \times 512$ for each $z = 1, \ldots, Z$. Furthermore, we divide the slices of size $512 \times 512$ into smaller patches with a predetermined stride. Patching big images into smaller pieces has two benefits: (i) it makes the input size smaller and more tractable and (ii) it helps to provide more training data, where there is a shortage of medical imaging data on hands. Figure 3 demonstrates the training flow.

## 4. Numerical Results

In this section, we provide numerical results to show the performance of the proposed U-Net model in approximating the PDD values. This numerical simulation study was granted an exemption from the Institutional Review Board.

From the 100 patients' data, we randomly selected 65 patients for training, 15 patients for the validation, and the remaining 20 patients for testing. As we have discussed in the previous section, the original slices of size $512 \times 512$ are divided into smaller patches. For this experiment we pick the patch size 128 with stride 64. To avoid over-fitting, a dropout regularization was added to the U-Net model with rate 0.1 after each convolution layer. The ADAM optimization algorithm with a learning rate of 0.001, and batch size 32 is used to update the model. These hyper-parameter numbers are obtained empirically as we get the best results with these values. The training was run on a server with two Nvidia Tesla V100 32 GB for 30 hours.

The U-Net model is trained over data related to head and neck cancer. We define the absolute and relative error for every element in tensor $512 \times 512 \times z$ as the following:

$$\texttt{Abs-Error} = |\texttt{dose proposed by PDD - dose proposed by U-Net}| \tag{1}$$

$$\texttt{Rel-Error} = \frac{|\texttt{dose proposed by PDD - dose proposed by U-Net}|}{\texttt{dose proposed by PDD}} \times 100 \tag{2}$$

When the U-Net model is trained, the errors defined in (1) and (2) are measured for all cancerous organs and OARs for all testing cases. The mean and standard deviation values (`mean+std`) are reported in Table 1. In order to obtain a better sense of the errors, we provide the average of PDD dose values in the Gray (Gy – the scale to measure the dosage) in the first row. Also, the Box plot for testing cases is represented in Fig 4. As it is shown, compared to the PDD values, DNN achieves a pretty small error on predicting the dose-prediction for the cancerous organs— an average of 2.23 absolute error resulting in average relative error of 3.23%. Similarly, for the OARs, DNN obtains pretty small errors, an average of 3.63 absolute error resulting in average relative error of 6.98%, although the results have higher variance than that of in cancerous organs. The reason is that OARs are affected by radiation differently for every single patient. This highly depends on the prescribed value for the nearby cancerous tissues, beam angles, and the patient anatomy itself. Therefore, data points for OARs are nosy indeed, which causes high variance predictions. Increasing the number of patients in the training would mitigate this behavior.

| | PTV-SR | PTV-HR | BrainStem | Cord | Larynx | Parotid-left | Parotid-right |
|---|---|---|---|---|---|---|---|
| PDD (Gy) | 53.97 | 61.55 | 50.32 | 16.65 | 30.50 | 18.34 | 28.07 |
| U-Net (Gy) | 51.08 | 58.51 | 48.12 | 14.38 | 30.55 | 16.74 | 26.88 |
| Abs-Error | $2.35 \pm 1.6$ | $2.12 \pm 1.7$ | $3.55 \pm 2.15$ | $4.05 \pm 2.11$ | $3.80 \pm 1.57$ | $3.60 \pm 1.98$ | $3.19 \pm 1.23$ |
| Rel-Error (%) | $3.78 \pm 2.6$ | $2.67 \pm 2.70$ | $8.21 \pm 4.30$ | $9.77 \pm 2.15$ | $9.83 \pm 2.68$ | $1.48 \pm 2.32$ | $5.62 \pm 2.54$ |

Table 1: The average relative error for cancerous organs and OARs.

For visualization purposes, we randomly picked one of the testing patients (results of more patients are reported in Appendix) and draw the heat map of dose distribution proposed by the U-Net model and PDD along with the contour of the associated slice. In Figure 5, every column is associated with a specific slice. As illustrated, the U-Net model closely imitates the PDD values, suggesting the performance of the trained model.

One common criterion for measuring the plan quality is the cumulative dose-volume histogram (DVH), which represents the percentage volume receiving greater than or equal to the value in the corresponding dose bin. In Fig 6, the obtained cumulative DVH by both PDD and U-Net is plotted for PTV-SR, PTV-HR, and OARs. From these plots, we can

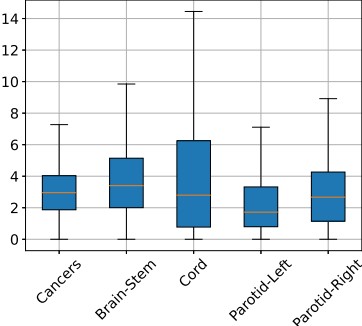

Figure 4: Box plot of the absolute error for cancerous organs and OARs.

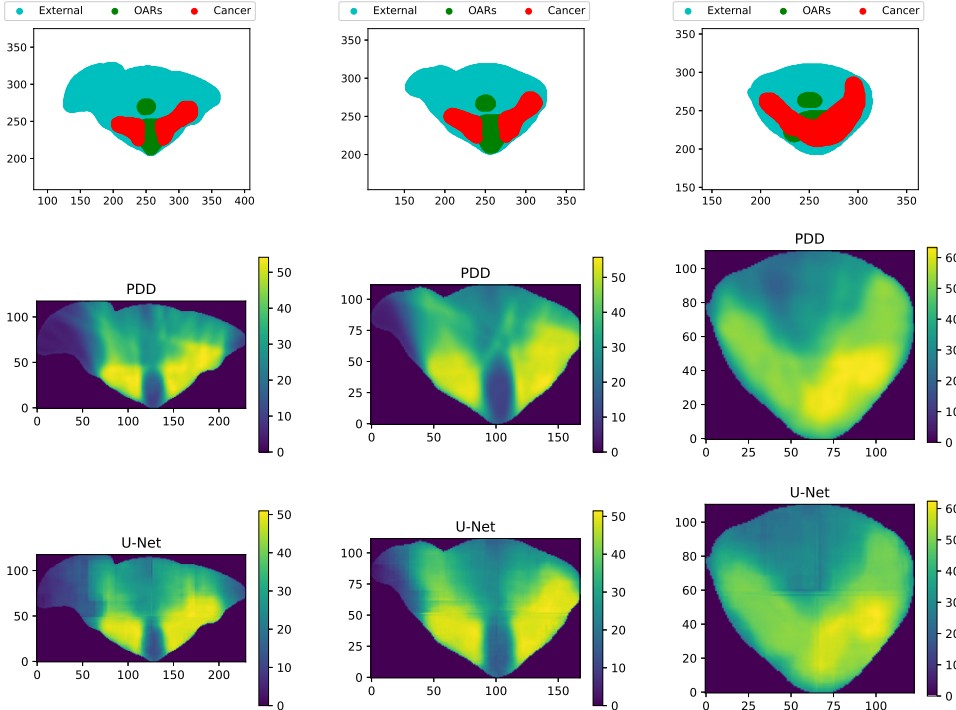

Figure 5: Each column shows one specific slice. The first row is the contour representing different structures. The second row and the third row are the heat map of the dose distribution proposed by PDD and U-Net model, respectively.

observe that U-Net mimics the PDD method very well, and there are very small deviations, especially in the cancerous organs. More DVH plots are plotted in Appendix.

As we have discussed in the Feature Engineering section, one of the attributes that we define for every structure is the dose limit value, which is (i) the prescribed value for cancerous structures, and (ii), either the average or the max limit value for OARs. Therefore, another important metric to evaluate the performances of the proposed model is to compare the dose limit violation imposed by this model with that one of the PDD. In Figure 7, a bar plot demonstrates the dose limit violation for OARs for both PDD and U-Net model.

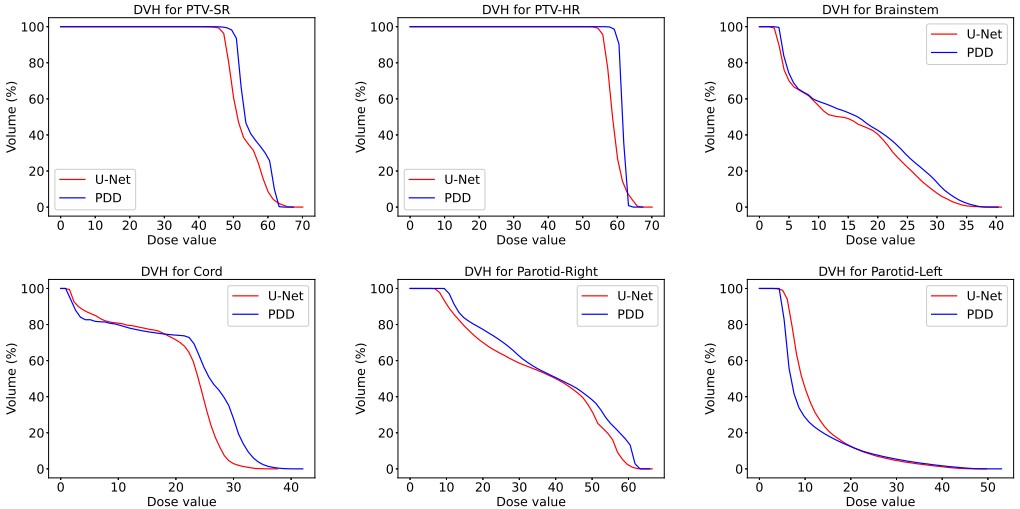

Figure 6: Cumulative DVH plot for a given structure illustrates the dose bins on x-axis versus the volume percentage of the structure receiving dose greater than to the bin on y-axis. In this figure, cumulative DVH is plotted for both PDD and U-Net. The closer the gap between these plots, the better the U-Net model is trained.

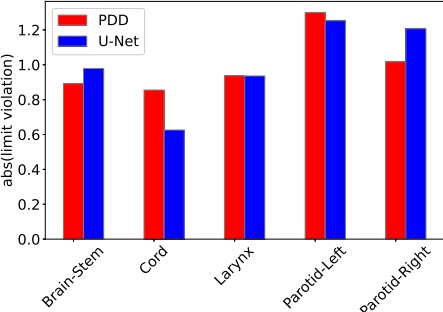

Figure 7: Bar plot indicating the average of limit violation over OARs through using each algorithm

Through comparing the bar heights for different structures it can be observed that the dose violation imposed by both methods are very close.

## 5. Conclusion

We trained a U-Net model, with specialized feature inputs to compute the 3D dose distribution for radiation therapy. To conduct the training, we utilized the existing treatment plans obtained from planned dose distribution, which is a combination of an optimization algorithm and interactive human planner inputs. The trained U-Net model provides the dose distribution for each patient very quickly, in less than a second. The numerical results indicate that U-Net provides dose values with high accuracy. In this work, we focused on obtaining the dose distribution which is one of the three main factors of the radiotherapy solution. Obtaining a fast solution for other components, i.e., *beams orientations* and *dynamic beam aperture shapes*, would be a future research direction.

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

## Appendix

In this section, we provide more numerical results for more testing cases. Figures 8 and 10 present the DVH plot for a patient, not seen in the training. In addition, Figures 9 and 11 illustrates the heat map of dose distribution proposed with both the U-Net model and PDD algorithms, and the contour of the associated slice. As it is shown, similar results like those shown in Figures 6 and 5. Although, we show more the heat map of dose distribution for more slices for each patient. Besides, we presented six slices of heat maps instead of three (as of those in Fig 5) to show the performance of our trained model on different parts of the body. As, Figures 9 and 11 suggests, the trained model works very well in organs close and far from the cancerous organ.

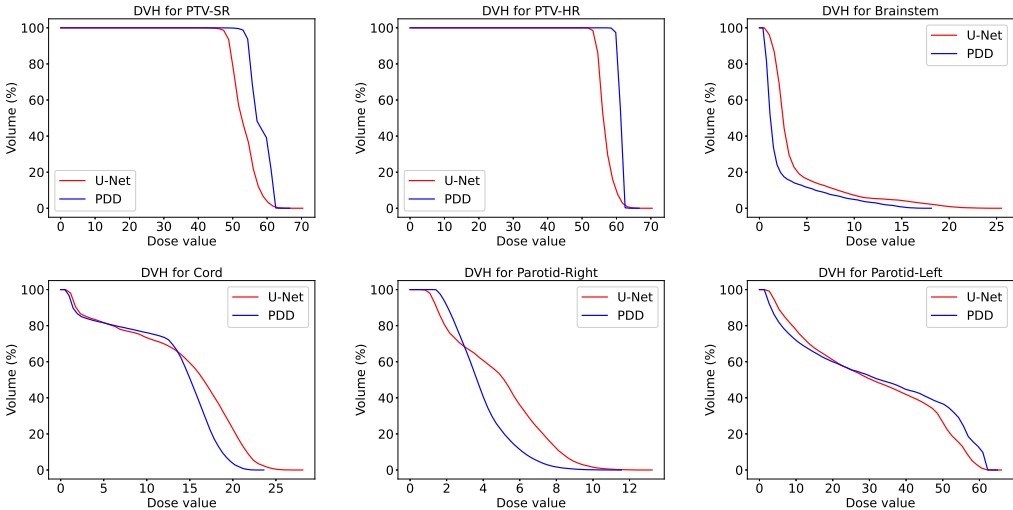

Figure 8: Cumulative DVH plot for a given structure illustrates the dose bins on the x-axis versus the volume percentage of the structure receiving dose greater than or equal to the bin on the y-axis. In this figure, cumulative DVH is plotted for both PDD and U-Net. The closer the gap between these plots, the better the U-Net model is trained.

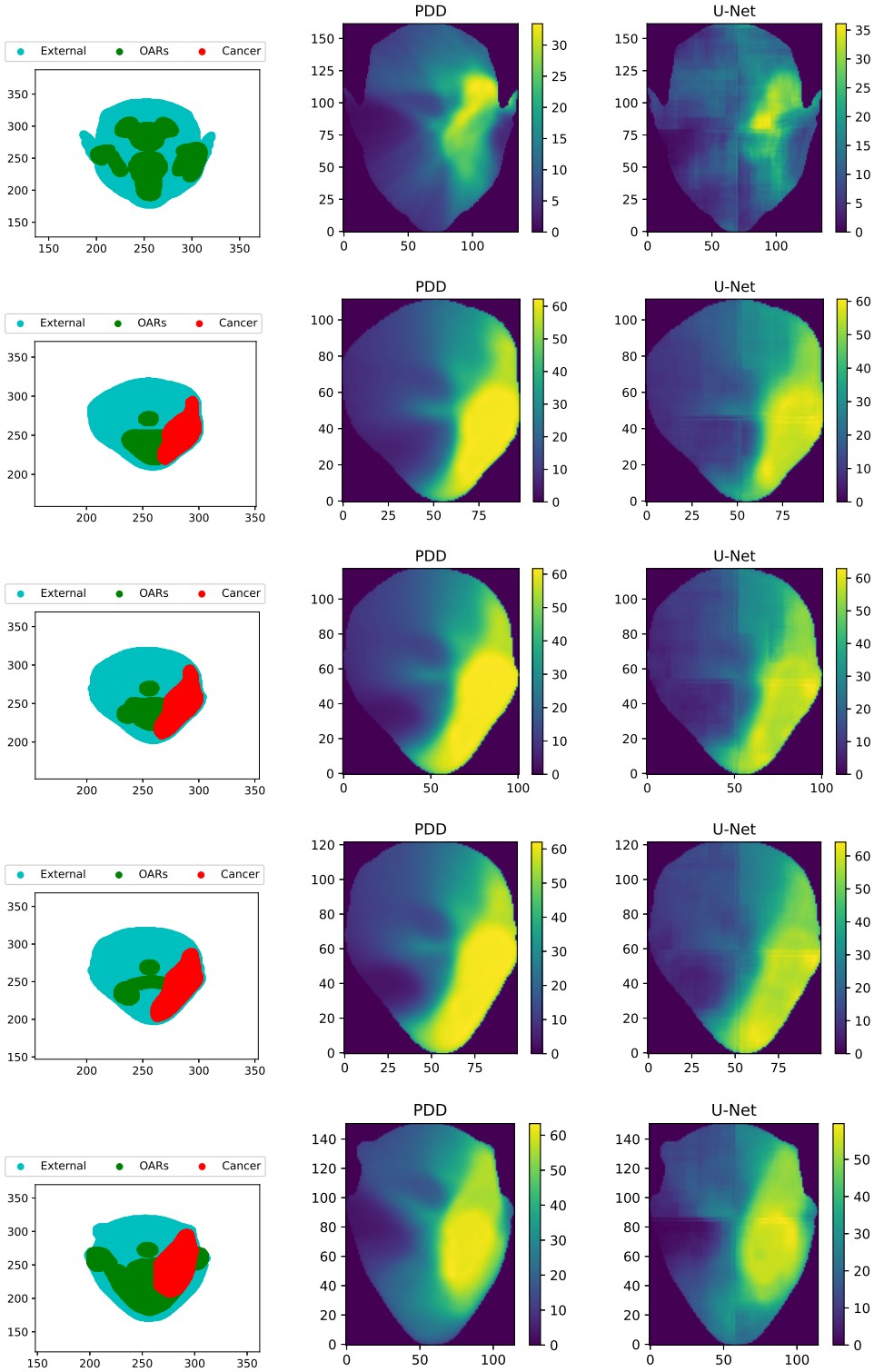

Figure 9: Each row shows one specific slice. The first column is the contour representing different structures. The second column and the third column are the heat map of the dose distribution proposed by PDD and U-Net model, respectively.

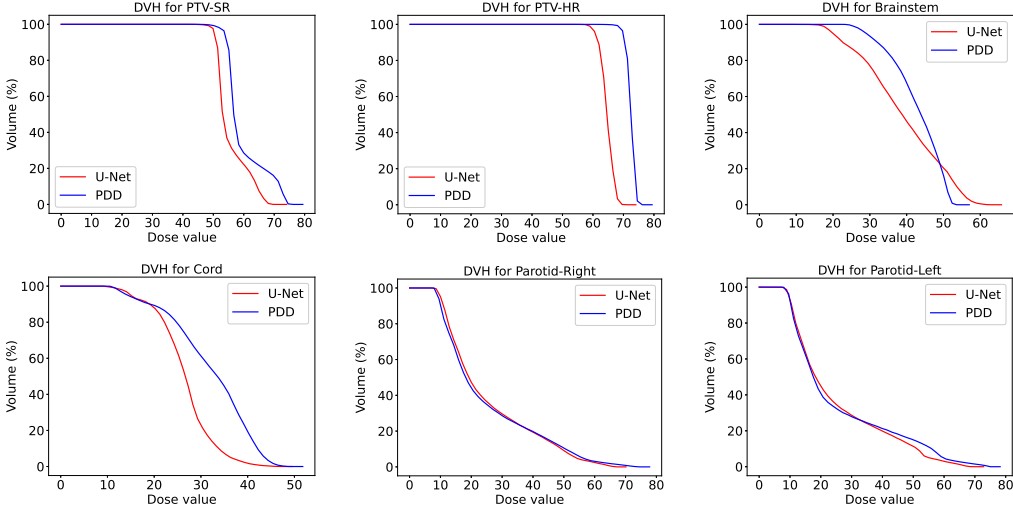

Figure 10: Cumulative DVH plot for a given structure illustrates the dose bins on the x-axis versus the volume percentage of the structure receiving dose greater than or equal to the bin on the y-axis. In this figure, cumulative DVH is plotted for both PDD and U-Net. The closer the gap between these plots, the better the U-Net model is trained.

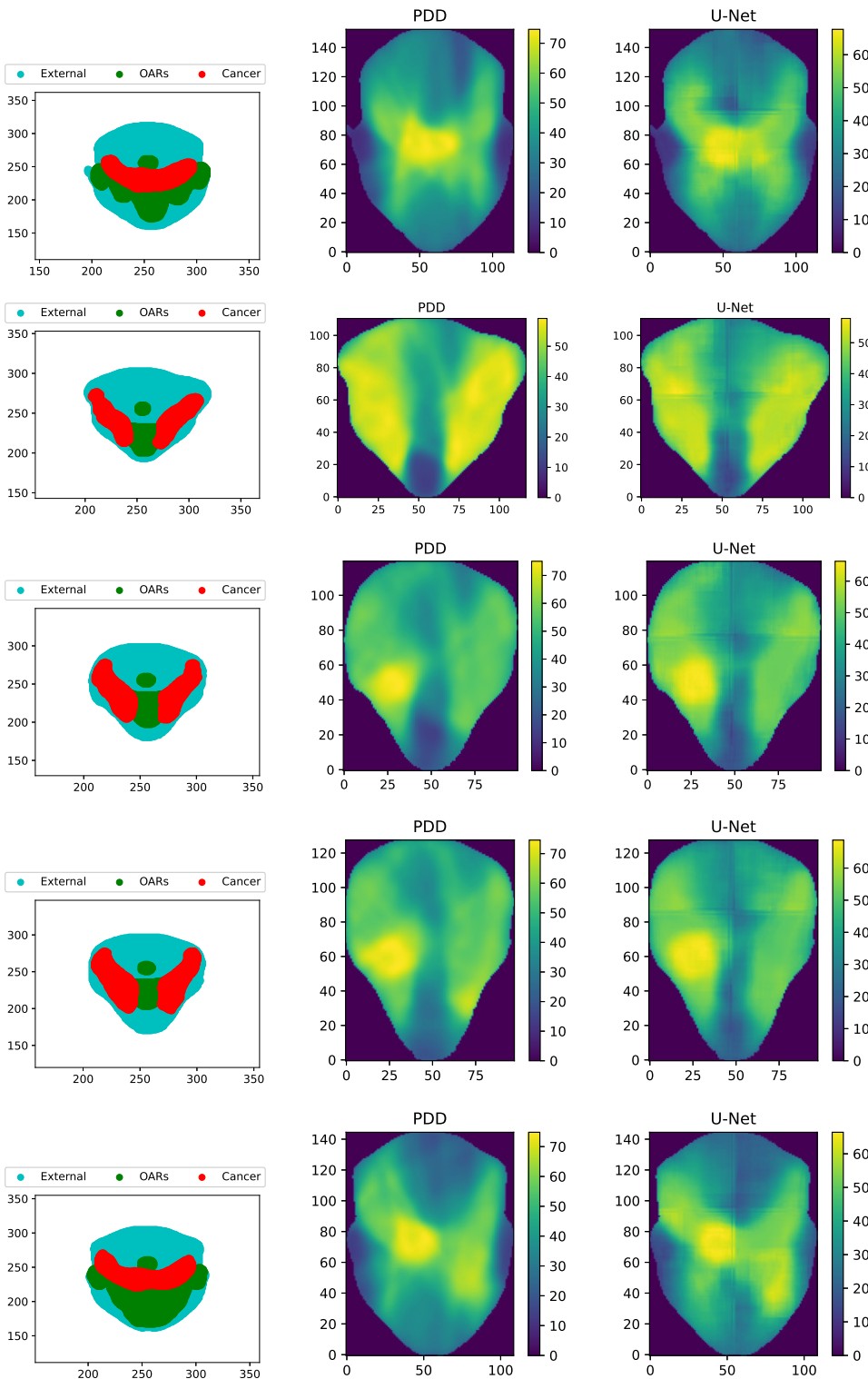

Figure 11: Each row shows one specific slice. The first column is the contour representing different structures. The second column and the third column are the heat map of the dose distribution proposed by PDD and U-Net model, respectively.

