# OpenReview forum: "Automated Learning of Radiation Therapy Dose Distribution for Head and Neck Cancer"
_MIDL.io/2021/Conference — Submitted to MIDL 2021_

### Official Review · AnonReviewer1 · 2021-03-05

**Confidence:** 5
**Preliminary Rating:** 2
**Final Rating:** 1

**Summary:**

The paper used a Unet to perform 3D dose prediction, given the prescription dose and the various organ constraints for head and neck cancers. The main advantage of this approach is an automatic approach for generating dose prediction at pixel level in a computationally fast manner. Standard treatment optimization would take several hours and potentially require user corrections.

**Strengths:**

An automatic approach to perform dose prediction is interesting and useful for both automating and generating consistent treatment planning solutions for radiation therapy. The proposed approach shows promising results and is applied on a reasonably sized dataset.

**Weaknesses:**

Its not clear how the various constraints are introduced for planning -- that is are the dose limits for all the organs and the target coverage enforced in the same footing? Doesn't this then create a problem because sometimes certain OARs are close to the target and in such cases it may not be possible to ensure dose a the prescribed limits while maintaining coverage. How is this handled by the approach? The presented results only showed the overall averages and errors compared to the planned dose description but did not look at how acceptable were the predicted treatment plans for clinical use. This is really important for applying these techniques in clinic, yes?


**Deanonymize Review:**

no

**Detailed Comments:**

The error plots Fig 7 is very useful to understand that the deep learning approach produces reasonable estimate. But one main concern is that the accuracy is measured averaging over the whole image. This presents a problem - because large deviations in regions that are not close to high dose may be OK but small errors close to the target may be problematic. How well does the method handle target coverage (no metrics for target coverage are shown here), or dose for the organs.

The cumulative DVH curves show large dose deviations for cord (albeit in Fig 6 this is towards lowering dose, but the dose to PTV is also reduced in the same case, is that acceptable?). Fig 8 shows even large deviations for PTV-SR and PTV-HR with lower dose using Unet vs. PDD, and much higher doses to cord, brain stem, and right parotid glands.

Because the paper really is an application of a well known technique to a new problem, it would be helpful to analyze how the technique helps to meet sufficient quality in the predicted treatments using this method using metrics for coverage, and fine grained metrics for organ dose.

Minor: Units for the dose maps - Is it Gy? Y-axis units for box-plot Fig 4 is missing.

**Final Rating Justification:**

There was no rebuttal or revision. Hence, I keep my recommendation to reject.

**Justification Of The Preliminary Rating:**

Although the qualitative results look reasonable, quantitative metrics to clearly show how good the predicted dose maps are for meeting the constraints for target coverage and dose constraints for organs are lacking. A more indepth study using finer grained metrics would be more instructive.

**Paper Type:**

validation/application paper

**Questions To Address In The Rebuttal:**

The main question is how good are the predicted treatment plans for ensuring target coverage and meeting dose constraints for organs? The average metrics using cumulative DVH plots are somewhat useful but how well are the constraints met is not answered. Without that its hard to know the real utility of the deep learning approach for this problem.

**Special Issue:**

no

---

### Official Review · ~Hans_Meine1 · 2021-03-08

**Confidence:** 4
**Preliminary Rating:** 2
**Final Rating:** 1

**Summary:**

The proposed approach directly estimates the (planned) dose distribution from a treatment plan (i.e., contour sets of risk and target structures, plus the respective dose constraints). It consists of a (slightly modified) U-net trained on carefully chosen input features (prescribed dose values, segmentation labels and distance to target). Evaluation is based on voxel-wise comparison of the predicted dose with the results of a treatment planning system (TPS) that take much longer to compute.

**Strengths:**

The paper reads well and is relatively easy to follow, given enough background knowledge on radiotherapy. The figures depicting results are helpful and an appendix is used to provide more results. The problem is relevant and the results look good enough to be useful.

**Weaknesses:**

I have not researched prediction of RT dose distributions myself yet, but I was surprised that no prior work on this was cited. Indeed, a quick google search brought up numerous apparently very related works. Furthermore, the authors themselves write that "[CNN and U-Net] have been used and provided promising results in [applications including] dose quantification, and radiation adaptation."

Ablation studies are mentioned, but their results are not included in the paper. However, it would've been interesting for the reader to see him-/herself how much difference the various input features make.

While the diagrams in Fig. 1 look nice, they're not helpful but really misleading – for instance, I spent a while pondering how the proposed workflow replaces the "Optimization" (TPS, presumably) completely. As far as I can see, the iterative loop (TPS / human expert – the double arrow reflecting the iterative part in 1a is not visible enough) would also have to happen with the DNN-based approach (1b), and would have to be followed by the final, more expensive optimization in order to get a proper treatment plan. (Furthermore, the probablility that the TPS has to be run twice will be reduced, but still be greater than zero.)

The example dose distributions show tiling artefacts and may benefit from a proper overlapping tile strategy (with "valid" convolutions, cf. Ronneberger's work).

**Deanonymize Review:**

yes

**Detailed Comments:**

The whole argument about "After training, this model is able to provide a more robust guidance for the human expert toward designing the final treatment plan." confused me, as already stated above under "Weaknesses" – I believe the description of how the RT workflow would look like with your DNN should be clarified.

What does "these methods suffer from huge feature manipulation" mean?

It is left unclear how the proposed approach "reduces the [human expert] variability" (as stated in the abstract).

The proposed speed-up (1 second instead of 3-6 hours) sounds exaggerated. From what I heard, modern TPS take less than one hour. 1 second sounds plausible (hopefully including the preparation of the feature maps from the "patient information"), but the speedup is large enough anyhow that one does not have to exaggerate.

The phrase "… PDD is available for 100 anonymized patients at the [UNC]" suggests to readers that this is publicly available data. If this is not the case, please re-phrase.

"This numerical simulation study was granted an exemption from the Institutional Review Board." should qualify the IRB – it is probably related to my comment about the data availability.

Maybe you find a better term than "patient information" for the RT structure sets + dose values constraints? ("Patient information" often refers to age, sex, weight, …)

The caption of Fig. 2 speaks about "up-convolution layers", but the architecture diagram depicts "unpooling" layers (and therefore also do not change the number of channels).

The evaluation is not sufficiently reproduced in the paper:
- Figure 4 shows helpful box plots of the absolute error, but why was the relative error not depicted?
- Figure 5 is only for one patient, and I assume the same holds for Fig. 6 and 7, but that should be stated.
- When you write "Figures 8 and 10 present the DVH plot for a patient, not seen in the training." in the appendix, that triggers the question if the figures that are part of the main text are from a patient whose data was used for training?

"the type of tissue from the set T" should probably include a label for the background, right?

Fitting an exponential function to approximate the incidental dose should be explained and could be one of the actual contributions of this paper (but no more information than this half sentence is given about it).

Patching big images does not "provide more training data" – for this task, each voxel is a training sample (contributing to the loss).

Was any kind of early stopping used? The manuscript states that the training ran for 30 hours – how was overfitting prevented and / or convergence ensured?

"Increasing the number of patients in the training would mitigate this behavior." is a bold statement that is not justified – maybe just write "could" instead of "would"?

Unusual words: "patient length" (size), "two main blocks, namely the contracting block and the expanding block" ("block" typically refers to a small unit of layers; in this case "pathway" or "subgraph" would be more appropriate, for instance).

Plural missing in some places, e.g.: "Several performance measurements *is* …", "plot" in Fig. 6 caption, KBP does not predict a single DVH of a patient (grammatically, that sentence is correct, but semantically not).

Grammar / word too much: "higher variance than that of in …"

Spelling: "nosy" -> "noisy"

The English in the Appendix needs more proofreading.

Page 15 is empty.

**Final Rating Justification:**

Unfortunately, I see no rebuttal or revision, so I vote to reject this paper.

In particular, it would've been important to me to clarify the relation of the contribution to previous work.

**Justification Of The Preliminary Rating:**

Before googling for 'deep neural network "dose distribution" ', I was about to say "weak accept", but after realizing that there are already many works on this topic (from several years), I think it would be crucial to put this work into relation with prior work. A quick look at a few other papers indicated potential differences, but this is something the authors should do, not reviewers or other readers.

**Paper Type:**

methodological development

**Special Issue:**

no

---

### Official Review · AnonReviewer4 · 2021-03-09

**Confidence:** 4
**Preliminary Rating:** 3
**Recommendation:** Oral, Poster
**Final Rating:** 2

**Summary:**

This paper proposes a U-net based model to predict Planned Dose Distribution(PDD) for radiotherapy. By using four different features as input, the models is trained to predict PDD for each slice of patient image. The performance of the proposed method is evaluated with clnically relevant metrics such as absolute/relative errors, cummulative dose-volume histogram.

**Strengths:**

- The motivation of the study is well presented.
- An extensive literature review is provided to understand related works.
- The data and network architecture used are clearly described.
- The performance of the proposed method is evaluated with clinically relevant metrics.

**Weaknesses:**

- The experiement was performed with only one random split, which is hardly reproducible.
- The analysis of failure cases which is ciritical for radiotherapy should be included.
- The 'patch-based' training has both pros and cons.

**Deanonymize Review:**

no

**Detailed Comments:**

- In Figure 1(a), it is better to show as loop than the bidirectional arrow to show the interativenss of existing workflow.
- In page 6, "OARs are nosy" -> "OARs are noisy"
- In Figure 5, the map of U-net has grid effect which seems the reason of patch-based training. Please mention and discuss this point.
- There is no experimental results for inference time using U-net, which is mentioned earlier as 'less than a second'

**Final Rating Justification:**

After referencing the comment from other reviewer, some important parts such as literature survey and comparision with existing methods are missing. I changed my rating to 'weak reject'.

**Justification Of The Preliminary Rating:**

The paper tackles clinically important problems with clear benefits of machine learning-based method. With reletively classic CNN network, the proposed method achieved high accuracy for predicting dose distribution for radiotherapy. Clinically relevant metrics were prodived to assess the performance of the proposed method with intuitive visuall illustration.

**Paper Type:**

methodological development

**Questions To Address In The Rebuttal:**

- Please justify the claim that the inference time using DL or ML is less than a second.
- Please analyze the effect of data split for the training and validation, testing.
- Please also analyze the effect of patch-based method for performance and predicted PDD map.

**Special Issue:**

no

---

### Official Review · AnonReviewer3 · 2021-03-09

**Confidence:** 5
**Preliminary Rating:** 1
**Recommendation:** Poster

**Summary:**

This study proposes a suitable feature matrix for H&N cancer and train a CNN model to mimic the planned dose distribution procedure. The procedure is fast (less than a second), and conventional U-Net was trained for this purpose. The main problem of this approach is that there is no technical innovation in this study, as the problem is addressed by several scientists already and also U-Net is not unique, it has been done before.

**Strengths:**

--The problem is an important problem, and motivation is strong, efficiency is an important criteria in therapy planning.
--Problem definition and most of the literature are given in a concise and clear way, informative for readers (though literature is not complete).


**Weaknesses:**

--Generalizability of the method is questionable. Nowadays, the field is going towards methodologies where they are tested over multiple datasets. This study does not have that. There are head neck data sets for this purpose, in the public data, it can be used to test robustness and generalizability of the data.

--lack of technical novelty. Authors failed to show how their method is different than available U-Net algorithms for the same purpose.

--lack of comparisons with  state of the art methods.

**Deanonymize Review:**

no

**Justification Of The Preliminary Rating:**

--  technical novelty is absent
-- no comparisons with the state of the art
-- generalizability and robustness tests are missing (since this is not a proof of concept study, and already existing methods there, it is expected that there is a rigorous evaluation etc)

**Paper Type:**

validation/application paper

**Questions To Address In The Rebuttal:**

--Generalizability of the method is questionable. Nowadays, the field is going towards methodologies where they are tested over multiple datasets. This study does not have that. There are head neck data sets for this purpose, in the public data, it can be used to test robustness and generalizability of the data.

--lack of technical novelty. Authors failed to show how their method is different than available U-Net algorithms for the same purpose.

--I guess the system is 2D while there are existing 3D dose distribution prediction methods in the field. Please clarify.

--No comparisons with the state of the art.

**Special Issue:**

no

---

### Meta-Review · Area_Chairs · 2021-03-31

**Recommendation:** Reject

**Metareview:**

this is a clear case all reviewers found several issues, primarily missing novelty and limited literature review, final ratings are all in favour of rejection

**Paper Type:**

validation/application paper

---

### Decision · Program_Chairs · 2021-03-31

Reject